# Chitosan-Coated Solid Lipid Nanoparticles as an Efficient Avenue for Boosted Biological Activities of *Aloe perryi*: Antioxidant, Antibacterial, and Anticancer Potential

**DOI:** 10.3390/molecules28083569

**Published:** 2023-04-19

**Authors:** Tahany Saleh Aldayel, Mohamed M. Badran, Abdullah H. Alomrani, Nora A. AlFaris, Jozaa Z. Altamimi, Ali S. Alqahtani, Fahd A. Nasr, Safina Ghaffar, Raha Orfali

**Affiliations:** 1Department of Health Sciences, Faculty of Health and Rehabilitation Sciences, Princess Nourah bint Abdulrahman University, Riyadh 11671, Saudi Arabia; tsaldayel@pnu.edu.sa; 2Department of Pharmaceutics, College of Pharmacy, King Saud University, Riyadh 11495, Saudi Arabia; aomrani@ksu.edu.sa; 3Nanobiotechnology Unit, College of Pharmacy, King Saud University, Riyadh 11495, Saudi Arabia; 4Department of Physical Sports Sciences, College of Education, Princess Nourah bint Abdulrahman University, Riyadh 11671, Saudi Arabia; naalfaris@pnu.edu.sa (N.A.A.); jzaltamimi@pnu.edu.sa (J.Z.A.); 5Department of Pharmacognosy, College of Pharmacy, King Saud University, Riyadh 11451, Saudi Arabia; alalqahtani@ksu.edu.sa (A.S.A.); fnasr@ksu.edu.sa (F.A.N.); sghafar.c@ksu.edu.sa (S.G.)

**Keywords:** *Aloe perryi*, chitosan-coated solid lipid nanoparticles, antioxidant, antibacterial, anticancer

## Abstract

*Aloe perryi* (ALP) is an herb that has several biological activities such as antioxidant, antibacterial, and antitumor effects and is frequently used to treat a wide range of illnesses. The activity of many compounds is augmented by loading them in nanocarriers. In this study, ALP-loaded nanosystems were developed to improve their biological activity. Among different nanocarriers, solid lipid nanoparticles (ALP-SLNs), chitosan nanoparticles (ALP-CSNPs), and CS-coated SLNs (C-ALP-SLNs) were explored. The particle size, polydispersity index (PDI), zeta potential, encapsulation efficiency, and release profile were evaluated. Scanning electron microscopy was used to see the nanoparticles’ morphology. Moreover, the possible biological properties of ALP were assessed and evaluated. ALP extract contained 187 mg GAE/g extract and 33 mg QE/g extract in terms of total phenolic and flavonoid content, respectively. The ALP-SLNs-F1 and ALP-SLNs-F2 showed particle sizes of 168.7 ± 3.1 and 138.4 ± 9.5 nm and the zeta potential values of −12.4 ± 0.6, and −15.8 ± 2.4 mV, respectively. However, C-ALP-SLNs-F1 and C-ALP-SLNs-F2 had particle sizes of 185.3 ± 5.5 and 173.6 ± 11.3 nm with zeta potential values of 11.3 ± 1.4 and 13.6 ± 1.1 mV, respectively. The particle size and zeta potential of ALP-CSNPs were 214.8 ± 6.6 nm and 27.8 ± 3.4 mV, respectively. All nanoparticles exhibited PDI < 0.3, indicating homogenous dispersions. The obtained formulations had EE% and DL% in the ranges of 65–82% and 2.8–5.2%, respectively. After 48 h, the in vitro ALP release rates from ALP-SLNs-F1, ALP-SLNs-F2, C-ALP-SLNs-F1, C-ALP-SLNs-F2, and ALP-CSNPs were 86%, 91%, 78%, 84%, and 74%, respectively. They were relatively stable with a minor particle size increase after one month of storage. C-ALP-SLNs-F2 exhibited the greatest antioxidant activity against DPPH radicals at 73.27%. C-ALP-SLNs-F2 demonstrated higher antibacterial activity based on MIC values of 25, 50, and 50 µg/mL for *P. aeruginosa*, *S. aureus*, and *E. coli*, respectively. In addition, C-ALP-SLNs-F2 showed potential anticancer activity against A549, LoVo, and MCF-7 cell lines with IC50 values of 11.42 ± 1.16, 16.97 ± 1.93, and 8.25 ± 0.44, respectively. The results indicate that C-ALP-SLNs-F2 may be promising nanocarriers for enhancing ALP-based medicines.

## 1. Introduction

Natural medicine has been used for thousands of years and is still a significant area of study for developing new drugs. Natural products contribute heavily to drug development due to their chemical, pharmacological, and biological properties to treat disease [1]. They have served as the primary source of medications used to maintain human health due to their potential anticancer, antimicrobial, and anti-inflammatory properties [2]. Recently, a large number of pharmaceuticals have come from natural sources such as plants.

*Aloe perryi* (ALP) is a plant that is widely used in traditional medicines in many countries, primarily due to its therapeutic properties [1]. ALP is considered a promising medicinal agent because it contains several phytochemicals such as glycosides, phenols, proteins, flavonoids, and phytosterols, which potentiate its biological activities [1,3]. In addition, ALP has unique compounds such as C-glycosylated chromones, chromones, anthrones, and proanthocyanidins (phenolic compounds), which offer potential antioxidant, antimicrobial, anticancer, and anti-inflammatory activities [3,4]. Thus, the effectiveness of ALP in the treatment of many diseases, including wound healing, diabetes, inflammation, and various tumors has been confirmed [1,5].

ALP extracts have been revealed to significantly inhibit the growth of numerous cancer cell lines [6]. However, herbal extracts and isolated phytoconstituents do not successfully translate their effects into the therapeutic field when used in traditional forms due to their limited solubility, permeability, poor stability, and bioavailability [7].

Therefore, nanotechnology was introduced to overcome these restrictions. The nanoscaled delivery systems have been adopted in various biomedical applications [8]. For instance, the low aqueous solubility of sirolimus and fenofibrate is improved when they are prepared in nanocrystal form. The nanoemulsion form of cyclosporine improved its immunosuppression activity. When compared to conventional doxorubicin, its liposomal form demonstrated a reduced risk of cardiotoxicity and an improved response rate [9]. Due to the benefits of the nanodrug delivery systems, researchers extended their work to investigate the impact of these systems on the therapeutic activity of natural products. In addition, nanotechnology is considered an eco-friendly synthesis method that can contribute to the development of targeted drugs for infectious diseases and cancer. Therefore, many eco-friendly methods for the preparation of nanoparticle systems from plants have been recommended because of their economical nature, low toxicity profile, biocompatibility in nature, and enhancement of the biological activity and bioavailability of the secondary metabolites for the biological properties [10,11,12]. Thus, this technology will have a role in smart cities for sustainable development, preserving the environment, improving health indicators, and enhancing the quality of health services to improve the quality of life and achieve sustainability [13].

Solid lipid nanoparticles (SLNs) and chitosan (CS) nanoparticles (CSNPs) as biocompatible and biodegradable systems have shown promising results in the treatment efficacy of natural materials [14]. SLNs are lipid-based drug delivery systems, which are usually prepared without organic solvent, so highly physically stable systems were obtained [15,16]. They have the potential to accommodate both hydrophilic and lipophilic molecules [15]. Cationic nanocarriers are attractive delivery systems due to their extended residence time and improved therapeutic success [14,17].

CS is a natural polysaccharide obtained through the partial deacetylation of chitin. CS has many potential uses in pharmaceutical fields due to its biosafety, biocompatibility, biodegradability, and biological activity [8]. Chitin is formed from N-acetyl-D-glucosamine units that are joined together by glycosidic (1,4) bonds to form the linear biopolymer [18]. It is the second most abundant polysaccharide in nature after cellulose, found in exoskeletons, cuticles, and fungi. The most common source of CS is from the shells of crustaceans used in the seafood processing industry, such as crab or shrimp [18]. CS has a polycationic, harmless, non-toxic, and biodegradable chemical structure that is relatively stable, and it is biocompatible with a wide range of organs, tissues, and cells [18]. CS has been used in a variety of products, including food additives, wastewater treatment agents, live cell encapsulation, enzyme immobilization, cholesterol-lowering medications, biomedical wound dressings, and drug excipients in the pharmaceutical industry. Various factors, such as acid–base concentration, incubation time and temperature, and particle size have been found to affect the physicochemical properties of CS. Fungal CS has a lot of benefits for biomedical applications and wound dressings due to its molecular properties and wound-healing capabilities [18]. It has a lower antigen effect and polycationic properties that make it soluble in physiological pH ranges. It can be utilized as a non-viral gene delivery system and potential drug carrier. CS was used in wound dressings in two ways. Firstly, CS is highly effective against dangerous microorganisms at low concentrations. Secondly, it was used as a hemostatic agent to promote blood clotting [18]. The products of CS have enormous economic potential in the biomedical/tissue engineering and food industries [18]. Therefore, numerous ways for functionalizing CS for these purposes have been developed. Unfortunately, chitosan’s commercial use is still fairly limited. CS has been used as a coating material to generate numerous cationic nanoparticles, including SLNs in the field of pharmaceutical nanotechnology [19]. Remarkably, CS-coated SLNs (C-SLNs) are reportedly very promising carriers for the delivery of a variety of natural drugs [19]. Thus, the therapeutic effectiveness of ALP may be improved by using CS-coated SLNs. Therefore, the C-SLNs and CSNPs could be efficiently attached to the biological membrane and enhance their biological activities of ALP [14]. Moreover, C-NPs have been thoroughly investigated by researchers in the fields of drug administration, cancer treatment, biological imaging, and diagnosis [17]. Many researchers concluded that natural product-loaded CS-coated SLNs exhibited significant biological activities such as antioxidant, antimicrobial, cytotoxicity, and anti-inflammatory [19,20]. However, to the best of our knowledge, there has been no study conducting on the antioxidant and biological activities of ALP after loading into SLNs, C-SLNs, and CSNPs.

In general, C-SLNs and CSNPs might be useful for enhancing ALP’s biological activity. Accordingly, the main objective of the current work was to generate and evaluate C-SLNs and CSNPs containing ALP extract. The obtained formulations were then examined for their particle size, zeta potential, stability, in vitro drug release, and morphology. The C-SLNs and CSNPs were assessed for potential uses based on their antioxidant and antimicrobial activity against *P. aeruginosa*, *S. aureus*, and *E. coli* as well as anticancer activity against A549, LoVo, and MCF-7 cell lines.

## 2. Results and Discussion

### 2.1. GC-MS of ALP Methanolic Extract

The GC-MS technique was performed to gain insight into the constituents of a volatile nature. This technique is very fast and accurate in determining volatile compounds in plants. Around 15 compounds of phytochemical substances were identified in ALP as shown in Table 1. Figure 1 shows the GC and GC-MS chromatograms of the methanolic extract of ALP. The following phytochemicals were confirmed: 3-amino-2-oxazolidinone, n(2)-isobutyryl-2-methylalaninamide, hydroxy dimethyl furanone, 2,3-dihydroxy-propanal,4-vinylphenol, 5-methyl-1,3-benzenediol, cinnamic acid, d-allose, 3-(4-hydroxy)-2-propenoic acid, 1,19-eicosadiene, 2, 9-octadecenoic acid, caprinitrile, 9-hexadecenoic acid, 1-hexyl-2-nitrocyclohexane, and docos-13-enoic acid.

### 2.2. The Content of Total Phenolic and Flavonoid of ALP Extract

Polyphenols and flavonoids are the two main types of AP phytochemicals that act as primary antioxidants and free radical scavengers [19]. These phenol-based antioxidants are useful modulators of redox homeostasis under oxidative stress, support the regulation of cellular activity, and lower the risk of chronic illness. Accordingly, it is essential to quantify their overall amount in plant extracts. The total phenolic content of ALP extract was 187.4 ± 10.36 mg/g GAE as determined using the Folin–Ciocalteu colorimetric technique. This suggests that ALP extract contains a high phenolic concentration (in GAE). Flavonoids are a type of plant compound that contain hydroxyl groups and have radical-scavenging capabilities. Numerous studies have found significant relations between flavonoid concentration and the inhibition of tumor growth as well as the activation of apoptosis in cancer cells [21]. Using an aluminum chloride-based colorimetric technique, the flavonoid concentration of ALP extract was measured to be 33.2 ± 4.11 mg of QE/g extract. This study result shows that ALP extract has high amounts of phenolic and flavonoid compounds, which are major contributors to antioxidative, antibacterial, and anticancer activities.

### 2.3. Characterization of ALP-SLNPs

#### 2.3.1. Physicochemical Characterization

SLNs have attracted a great deal of interest as important nanocarriers in the field of drug delivery. Compritol 888 ATO is one of the most common solid lipids used in a variety of pharmaceutical delivery systems. It is generally considered to be safe (GRAS) and is commonly used in the production of SLNs [22,23]. In addition, the stabilizers were added to SLNs during production to enhance the amount of encapsulated drug while decreasing particle size. Surfactants (stabilizers) may modify the surface of SLNs, causing the prevention of particle aggregation. This behavior could produce a stable system [24]. Thus, using a surfactant is a better excipient to reduce particle size and maximize drug loading [25]. Consequently, proper surfactant selection is essential for producing effective SLNs [25]. Several surfactants have been explored in previous studies to achieve the desired SLNPs. TW80, a non-ionic surfactant, was broadly utilized for the development of SLNs in this study (Table 2). It has been reported that the presence of TW80 could modify the diffusion of the water/oil interface of the colloidal suspension, causing a change in the particle size of the SLNs [25]. In addition, CS-coated SLNs (C-SLNs) and CSNPs were developed due to their great properties such as positive charge, biocompatibility, bioadhesion, and cellular uptake. This study aimed to examine the SLNs, C-SLNs, and CSNPs based on particle size, EE%, drug release behavior, cytotoxicity, and antibacterial activity of ALP. The investigated formulations showed nanosized particles with a low PDI (<0.5). The average particle sizes of ALP-SLNs-F1 and ALP-SLNs-F2 were 169 and 138 nm, respectively (Table 2). The reduction in particle size in ALP-SLNs-F2 could be attributed to the presence of TW80, which has good emulsification properties [26]. TW80 is expected to adsorb at the oil/water interface, covering the oil phase and lowering the interfacial tension between the oil and water phases, resulting in a smaller particle size [27]. The smaller particle size of TW80 could also be ascribed to its high HLB value and minimized surface energy [15].

CS is a natural cationic polysaccharide that offers various advantages, including high membrane permeability, bioadhesiveness, and low toxicity [15]. The CS-coated SLNs may improve the biological activity of loaded extract [8]. Electrostatic interaction between CS amino groups and the lipid component of SLNs causes CS coating to develop [28]. The particle size of C-SLNs was elevated as shown in Table 2. The particle sizes of C-ALP-SLNs-F1 and C-ALP-SLNs-F2 were 185 and 177 nm, respectively. The presence of CS on the surface of SLNs could result in the formation of a viscous suspension, resulting in a larger particle size [15]. The particle size distributions of the resulting formulations are represented by PDI values less than 0.5, indicating that all formulations have homogeneous size distributions. There was a relationship between the PDI and particle size. This indicates a homogeneous system with a reasonable particle size distribution.

The zeta potential values of ALP-SLNs-F1 and ALP-SLNs-F2 were −12 and −16 mV, respectively. The particles tended to be more negatively charged after the addition of surfactant [29]. The zeta potential values of C-ALP-SLNs-F1 and C-ALP-SLNs-F2 were 11 and 14 mV, respectively. Due to electrostatic interactions, the CS-coated SLNs resulted in a formulation with positively charged SLNs, as expected. The successful coating is consistent with the conversion of the surface charge from the negative charge of SLNs to the positive charge after adding CS was confirmed [30]. The mean particle size and zeta potential of ALP-CSNPs were 215 nm, 28 mV, which increased significantly compared to the abovementioned formulation with a high positive value of zeta potential [8]. The increased particle size and surface charge inversion could confirm the CS on the surface of SLNs. These findings were in line with several research projects aimed at creating a lipid carrier with a CS layer [15,16]. It has been reported that C-SLNs are able to deliver therapeutic agents directly to the target cells, making them promising candidates for the delivery of ALP [19]. The interaction between positively charged SLNs and negatively charged cell membrane components could increase their benefits [28]. Additionally, the small particle size could also enhance endocytosis, which will increase cell internalization.

#### 2.3.2. Encapsulation Efficiency and Loading Capacity

Compared to solid lipids, Compritol 888 ATO is more efficient by high encapsulation efficiency, offering a greater area for drug loading because of its less perfect state [22]. The long chain length of behenic acid in Compritol 888 ATO increases drug intermolecular trapping via interchain intercalation [22]. TW80 stabilizer has a meaningful effect on the characteristics and encapsulation efficiency of SLNs [15]. Additionally, TW80 may have a high impact to reduce nanoparticle size by decreasing interfacial tension and increasing EE% and LC% [31]. The EE% and LC% of ALP in the several SLNs are displayed in Table 2. The EE% values for ALP-SLNs-F1 and ALP-SLNs-F2 were 67.4 ± 2.4 and 82.4 ± 1.7, respectively, and the values of LC% were 3.5 ± 0.3 and 5.2 ± 0.1 (Table 2). It has been revealed that the addition of TW80 to SLN composition causes an increase in the EE% and LC%. This influence could also be attributed to the aqueous phase’s micelle formation [31]. Furthermore, the EE% values for C-ALP-SLNs-F1 and C-ALP-SLNs-F2 were 64.6 ± 3.94 and 80.2 ± 5.3, respectively, and the LC% values were 2.8 ± 0.1 and 4.7 ± 0.1 (Table 2). The reduction in the LC% of CS-SLN samples is due to the increase in the total weight of these systems by the addition of the CS coat. Additionally, ALP-CSNPs had a mean size of 214.8 ± 6.6 nm, a positive zeta potential of 27.8 ± 3.4 mV, EE% of 76.4 ± 7.6, and LC% of 3.0 ± 0.3. The larger particle size of ALP-CSNPs was caused by the greater accessibility of protonated amine groups for ionic gelation of CS [8]. The reasonable EE% and DL% provided in the current study may be ascribed to the appropriate choice of the surfactant type and solid lipid.

#### 2.3.3. In Vitro Release Study

In vitro drug release profiles of ALP-loaded SLNs, CS-coated SLNs, and CSNPs are demonstrated in Figure 2. ALP solution in methanol was entirely released within 4 h. However, SLN release of ALP was in a biphasic pattern, with a high-release phase at 6 h and a slow-release phase at up to 48 h. In the first 6 h, approximately 45%, 60%, 38%, 43%, and 33% of ALP were released from ALP-SLNs-F1, ALP-SLNs-F2, C-ALP-SLNs-F1, C-ALP-SLNs-F2, and ALP-CSNPs, respectively. In addition, after 48 h, 86%, 91%, 78%, 84%, and 74% were released from ALP-SLNs-F1, ALP-SLNs-F2, C-ALP-SLNs-F1, C-ALP-SLNs-F2, and ALP-CSNPs, respectively. The small particle size of ALP-SLNs-F2 and the presence of TW80 could explain the highest drug release [32]. There was a statistically significant difference in the percentage of ALP released from coated and uncoated formulations, which may be caused by the existence of a CS layer, as shown by low EE% values. The biphasic release profiles of the test formulations point to the distribution of ALP inside and on the outer surface of the particles. The localization of ALP on the surfaces of SLNs during nanoparticle crystallization might be considered for a high-release phase [16]. The phase of sustained release was due to the ALP diffusion from the particle core to the receptor fluid from the aforementioned formulations [16]. Furthermore, the CS-coated NPs exhibited a lower burst and sustained release which could be attributed to the presence of a CS layer that hinders ALP release [31]. Despite the hindering effect of CS coating, the C-NPs were designed to augment the biological activity of many drugs [15]. Furthermore, due to CS deprotonation, an initial rapid release followed by a low release was observed for ALP-CSNPs, resulting in improved ALP diffusion [8]. The current findings agree with previous research on the release of rifampicin from SLNs and C-SLNs by Vieira et al. [30]. These authors demonstrated that C-SLNs had a slower RIF release profile than uncoated ones. These findings suggest that the majority of the drug remained in the C-SLNs following exposure to the biological conditions and was subsequently released into the targeted cells.

As shown in Table 3, the in vitro release kinetic model of ALP from the formulated systems was estimated using various mathematical models, including the zero-order, first-order, Higuchi, and Korsmeyer–Peppas kinetic models. The R^2^ value of the correlation coefficient was used to determine the kinetic model of ALP, in which the mathematical model with the highest R^2^ value most likely represents the release kinetic model. Accordingly, the data presented in Table 3 and ALP release model of C-ALP-SLNs-F1, C-ALP-SLNs-F2, and ALP-CSNs fit with Korsmeyer–Peppas’s equation. As shown, the n value (release exponent) is used to describe various release mechanisms. It was discovered that the majority of n values are less than 0.43, implying that the release mechanism was controlled by diffusion. This behavior proposes that the released drug from these carriers obeys a Fickian transport pattern. Table 3 demonstrates that the models of the release of ALP from ALP-SLNs-F1 and ALP-SLNs-F2 fit with the first-order equation. This pattern suggests that the drug released from the system is proportional to the concentration of ALP.

The current findings are in agreement with a previous study that has demonstrated a comparable pattern, suggesting that the results may be reliable and reproducible [33].

#### 2.3.4. Morphology of ALP-SLNs-F2 and C-ALP-SLNs-F2

SEM images of ALP-SLNs-F2 showed a smoothly spherical shape with consistently distributed nanoparticles (Figure 3). In TEM images, C-ALP-SLNs-F2 showed a relative increase in size with a rough surface when compared to a smooth surface of ALP-SLNs-F2. The images of coated and uncoated SLNs show no ALP crystals, indicating that the nanoparticles have a high ALP loading.

However, the particle sizes were slightly larger than those obtained from DLS due to the lyophilization process. The size of ALP-SLNs-F2 was smaller than that of C-ALP-SLNs-F2, possibly due to the successful coating of CS, which increases the diameter of NPs [33].

#### 2.3.5. Physical Stability Study

The obtained formulations were relatively stable with a minor increase in particle size (Table 4) and remained within an acceptable range after storage for one month. No signs of ALP precipitation during storage were observed. This suggests a homogeneous system with a suitable particle size distribution. There is no significant change in zeta potential values after one month of storage. %EE values did not demonstrate any significant change after a month.

### 2.4. Biological Activities of ALP-Loaded SLNs, C-SLNs, and CSNPs

#### 2.4.1. Antioxidant Activity

ALP-SLNs-F2, C-ALP-SLNs-F2, and ALP-CSNPs showed the highest EE% and ALP release, therefore, they were selected to explore the biological activity of ALP. Reactive oxygen species are known to cause DNA damage and lipid peroxidation, which are two conditions that are linked to cancer and infectious disorders [8]. The anticancer and antibacterial activities of ALP extract may be attributed to its antioxidant activity. Thus, the antioxidant activity of ALP-loaded NPs was assessed by measuring the scavenging activity towards DPPH. DPPH assay is the most common method to evaluate the effectiveness of antioxidants at scavenging free radicals [34]. The antioxidant activity of ALP-SLNs, C-ALP-SLNs, and ALP-CSNPs was assessed based on their concentration (Figure 4). The results indicated that when ALP extract concentration increased, the ALP extract’s antioxidant activity also increased. The antioxidant activities were concentration-dependent, and ALP-SLNs-F2 and C-ALP-SLNs-F2 showed higher antioxidant activity than that observed with blank NPs using the same concentration. This indicated that the formulation had an important impact based on the extract’s antioxidant activity. The evaluation of the DPPH results revealed an increase in antioxidant activity dependent on the concentrations [35]. ALP extract showed antioxidant activity against DPPH radicals of 61.30% at 100 µg/mL. C-ALP-SLNs-F2 showed the considerably greatest antioxidant activity against DPPH radicals at 73.27% followed by ALP-CSNPs at 68.61% at 100 µg/mL. The IC50 values of ALP-SLNs-F1, ALP-SLNs-F2, C-ALP-SLNs-F1, C-ALP-SLNs-F2, and ALP-CSNPs were 73, 66, 61, 45, and 58 µg/mL, respectively. Ascorbic acid was used as a standard and showed an IC50 value of 16 µg/mL. The IC50 value of ALP extract is 68 µg/mL. The remarkable DPPH-scavenging activity of ALP-loaded NPs and ALP methanolic extract results from the neutralization of free radicals by the transfer of hydrogen or an electron [8]. It has been reported that the antioxidant activity of aloe vera was enhanced when incorporated with cationic nanoparticles [34].

#### 2.4.2. Antibacterial Activity

Previous studies reported that the herbal extract-loaded C-NPs have great potential for utilization in many medical applications [15,36]. The comparisons of in vitro antibacterial activity of the ALP extract and ALP-loaded SLNs, C-SLNs, and CSNPs with blank NP were evaluated using microdilution broth antibacterial assay. Antimicrobial activities against *S. aureus* (Gram-positive bacterium), *P. aeruginosa*, and *E. coli* (Gram-negative bacteria) were explored. All of the examined microorganisms were more susceptible to the ALP-loaded NPs than the ALP extract alone (Figure 5). ALP extract and plain NPs showed no antibacterial activity at 200 µg mL^−1^. All ALP-loaded NPs showed antibacterial activity at 200 ug mL^−1^ against *S. aureus*, *P. aeruginosa*, and *E. coli*. Figure 5 displays the antibacterial activity of ALP extract, ALP-SLNs-F2, C-ALP-SLNs-F2, and ALP-CSNPs. The methanolic extract of ALP demonstrated effective antibacterial activity, demonstrating the value of methanol as an organic solvent for extracting ALP. ALP-CSNPs and C-ALP-SLNs-F2 were found to have potential antibacterial activity in all tested bacterial strains. Accordingly, ALP-CSNPs exhibited the maximum activity against *S. aureus* and *E. coli*, which indicates that they are more effective against common infections. Moreover, C-ALP-SLNs-F2 displayed the maximum activity against *P. aeruginosa*. Thus, higher antibacterial activity was shown by C-NPs and CSNPs compared to that observed in blank NPs (Figure 5). On the other hand, the blank CS-coated NPs had a noticeable effect on the studied bacterial strains (Figure 5). The MIC of ALP-SLNs-F2 was 100 µg/mL for all tested bacteria. In addition, the MIC values of C-ALP-SLNs-F2 were 25, 50, and 50 µg/mL for *P. aeruginosa*, *S. aureus*, and *E. coli*, respectively. However, the MIC values of ALP-CSNPs were 50, 25, and 25 µg/mL for *P. aeruginosa*, *S. aureus*, and *E. coli*, respectively. These results demonstrated the synergistic antimicrobial efficacy of ALP and CS-coated NPs. These results are in agreement with the previous work on mint extract-loaded CSNPs that increased their antibacterial activity [37]. Additionally, Achillea millefolium-loaded CSNPs showed high efficacy against *B. subtilis* and *P. aeruginosa* compared to the extract alone [38]. This effect could be explained by the small size and positive surface charge of the particle. The electrostatic interactions between the positively charged amino groups of CS and negatively charged bacterial cell membranes may be the cause of the antibacterial effect [8]. Plant extract-loaded CS-coated nanoparticles have been shown to have the potential to be effective antimicrobial agents against a variety of microorganisms [19]. The results indicated the potential use of these nanoparticles in the food preservation and medical industries by showing encouraging results in inhibiting the growth of pathogenic bacteria. Therefore, the utilization of CSNPs or CS-coated NPs as carriers for ALP would hold significant potential of antimicrobial activity.

#### 2.4.3. Anticancer Activity

ALP exhibits anticancer action on several cancer cell lines according to earlier investigations [1]. The antiproliferative activities of ALP were proposed to be due to cell cycle arrest [7]. The anticancer activity of ALP-NPs was studied using lung carcinoma cells (A549), colon cancer cells (LoVo), and breast cancer cells (MCF-7). The cells were treated for 48 h with C-ALP-SLNs-F2, and ALP-CSNPs at various concentrations in comparison with doxorubicin (Figure 6). In the cancer cell lines, the cytotoxic effects of ALP extract solution, plain SLNs, and CS-coated SLNs were examined. Cells were treated with ALP and NPs (concentrations equivalent to 0–25 µg/mL ALP) for 48 h at 37 °C. The formulations demonstrated a dose-dependent reduction in cell viability (Figure 6).

Anticancer experiments showed that ALP-loaded C-ALP-SLNs-F2 was more effective than the ALP extract alone on the tested cancer cell lines, especially at concentrations of 25 µg/mL. C-ALP-SLNs-F2 showed a promising anticancer activity compared to CSNPs. It has been observed that ALP-CSNPs did not reach IC50 within the used range of ALP concentrations. In addition, the IC50 values of C-ALP-SLNs-F2 were 11.42 ± 1.16, 16.97 ± 1.93, and 8.25 ± 0.44, for A549, LoVo, and MCF-7 cell lines, respectively. Furthermore, ALP extract had IC50 values of 23.16 ± 1.25, 23.8 ± 1.15, and 9.63 ± 0.25 µg/mL, for A549, LoVo, and MCF-7 cell lines, respectively. C-ALP-SLNs-F2 had considerable cytotoxic activity, indicating that this formulation has potential cytotoxic activities against several cancer cell lines that demand further study. Previous studies showed similar results relating to the synergistic anticancer activity of plant extracts loaded into cationic nanoparticles compared to plant extract [19,33]. These cationic nanoparticles could be a promising alternative to cancer treatment. Further research is needed to fully understand the potential of this approach.

## 3. Material and Methods

### 3.1. Materials

Compritol 888 ATO was purchased from Gattefosse, 36 Chemin de Genas (Saint-Priest, France). Low molecular weight chitosan (96% degree of deacetylation) was purchased from Sigma-Aldrich Chemie, GmbH (Steinheim, Germany). Tween 80 was purchased from BASF (Ludwigshafen, Germany). All reagents and chemicals used for chromatography were of ultra-performance liquid chromatography grade. All other chemicals and organic solvents were of reagent grade.

### 3.2. Plant Collection and Extraction

Dried portions of resin of *Aloe perryi* (ALP) Baker (*Saber Socotri*) were purchased from the local market in 2022. The ALP was recognized by the Pharmacognosy Department, Faculty of Pharmacy, Sana’a University, Yemen (voucher specimen: Mo-Sq9). Dried ALP was crushed, powdered, and stored in an airtight container. The dry powder of ALP (50 g) was macerated with absolute methanol (100 mL) for 48 h followed by sonication for 30 min at room temperature, and the extraction process was repeated twice with 100 mL of methanol. Using a rotary evaporator at 60 °C, the collected supernatant was dried under evaporation (Rotavapor Model 011, Buchi, Flawil, Switzerland). The dried extract was stored for further use. After being dissolved in methanol, the extract was filtered through a 0.22 Pa TFE syringe filter (Fisher Scientific, Pittsburgh, PA, USA) for spectrophotometric assay. Three replicates of each sample were tested. The yield was determined with the equation below:(1)$$\mathrm{Yield}\;\left(\%\right)=\frac{W1}{W2}\times 100$$
where W1 is the weight of the extract after the evaporation and W2 is the dry weight of the sample.

### 3.3. Gas Chromatography and Mass Spectrometry (GC-MS) Analysis

The chemical components of the ALP methanolic extract were verified using GC-MS (Turbomass, PerkinElmer, Waltham, MA, USA). The temperature program was started at 40 °C, hold for 2 min, and then increased to 200 °C at a rate of 5 °C/min, then hold for an additional 2 min. At 5 °C/min, the temperature was raised from 200 °C to 300 °C and maintained for an additional 2 min. An online library was then used to locate the ALP components.

### 3.4. Determination of the Total Phenolic and Flavonoid Content of AP Extract

The total phenolic content of the ALP extract was measured using the Folin–Ciocalteu colorimetric test [8,39]. In particular, 200 µL of the Folin–Ciocalteu phenolic reagent was thoroughly combined with 500 µL of 2.0 *w*/*v*% ALP extract. Then, 2.5 mL of a 10% (*w*/*v*) Na_2_CO_3_ aqueous solution was added, and the mixture was stored in the dark for 30 min. The absorbance was measured at 765 nm with a UV–visible spectrophotometer (Thermo Scientific GENESYS 10S UV-VIS, Madison, WI, USA). The total phenolic content was determined and represented as milligrams of gallic acid equivalent (GAE) per gram of extract. To create a calibration curve, gallic acid was used as the standard. The total flavonoid content of the AP extract was measured using the aluminum chloride colorimetric method.

The aluminum chloride colorimetric technique was used to determine the total flavonoid concentration of the AP extract [39]. Briefly, 0.5 mL of 5% NaNO_2_ and 0.5 mL of an ALP extract were mixed. Then, 1 mL of 10% AlCl_3_ was included. After that, the solution was treated with 2 mL of 1 M NaOH and allowed to sit for 30 min. The absorbance at 415 nm was measured using a UV–visible spectrophotometer (Thermo Scientific GENESYS 10S UV-VIS, Madison, WI, USA). A standard calibration curve was created using known amounts of a prepared quercetin solution. The calibration curve was used to calculate the total flavonoid content in the extracted sample, which was expressed as milligrams of quercetin equivalent (QE) per gram of extract.

### 3.5. Preparation of ALP-SLNs, C-ALP-SLNs, and ALP-CSNPs

ALP-SLNs were prepared by the probe ultrasonic melt-emulsification technique [15,40]. The composition of the SLNs is shown in Table 5. ALP extract (0.1% *w*/*w*) was dissolved in the melted lipid phase (mixture of Compritol 888 ATO and PL) which was heated to 85 °C using a water bath. The aqueous phase containing TW80 (2% *w*/*w*) was also heated to 85 °C using a water bath. Then, the heated aqueous phase (3 mL) was added dropwise to the melted lipid phase under ultra-sonication for 5 min with an amplitude of 60% to prepare pre-nanoemulsion (Sonoplus HD 220; Bandelen, Berlin, Germany). The obtained o/w pre-nanoemulsion was then added to 7 mL of cold water and kept at room temperature for 4 h for equilibration. Finally, the dispersion was further ultra-sonicated for 5 min with an amplitude of 60% using an ice bath to create ALP-SLNs. The same procedure was used to prepare blank SLNs without ALP extract. The dispersion was centrifuged at a slow rate (3000 rpm) for 5 min to remove undissolved ALP and large aggregated particles. The influence of different surfactants on the particle size and APL loading was studied (Table 1). Each formulation was produced in triplicate.

C-ALP-SLNs were prepared by coating the prepared ALP-SLNs with CS using a gentle physical adsorption method [30]. Briefly, CS solution (1% *w*/*v* in 0.2% *v*/*v* of acetic acid) was prepared and filtered through 0.42 m to remove any aggregates. The prepared chitosan solution was added to the ALP-SLNs or SLNs with gentle continuous shaking for 24 h at room temperature to obtain C-ALP-SLNs.

ALP-CSNPs were prepared to utilize the ionic gelation method as reported previously [8], with some modifications. Sodium tripolyphosphate solution (0.5 mg/mL) was added in a dropwise manner to CS solution containing ALP extract in a ratio of 1:2 under continuous stirring for 4 h. ALP-CSNPs were further ultra-sonicated for 5 min with an amplitude of 60% using an ice bath to create ALP-CSNPs. The final concentration of CS was 1 mg/mL.

### 3.6. Characterization of ALP-SLNs

#### 3.6.1. UV–Visible Spectrophotometry

The wavelengths of maximal absorption of ALP extract and plain SLNs, C-SLNs, and CSNPs were detected. They were serially diluted at various concentrations (2–12.5 µg/mL). A UV–visible spectrophotometer was then used to calculate the absorbance values of these samples (Thermo Scientific GENESYS 10S UV-VIS, Madison, WI, USA), and a calibration curve was plotted to check the linearity.

#### 3.6.2. Measurement of Particle Size, Polydispersity Index (PDI), and Zeta Potential

The mean particle size, PDI, and zeta potential of the obtained formulations were determined using a Zetasizer Nano ZS (Malvern Instruments, Worcestershire, UK). Dynamic light scattering (DLS) and laser Doppler velocimetry (LDV) modes were used to evaluate particle size and zeta potential, respectively, at 25 °C. The prepared formulations (10 µL) were diluted to 1 mL with deionized water, placed in a quartz cuvette, and examined at a scattering angle of 90°. All experiments were carried out in triplicate.

#### 3.6.3. Encapsulation Efficiency (EE) and Loading Capacity (LC)

The encapsulated amount of ALP within the formulations was determined as described previously [8]. Briefly, 1 mL of nanoparticle dispersion was ultra-centrifuged at 40,000 rpm for 30 min (Beckman Coulter, Pasadena, CA, USA) Then, the free ALP (supernatant) was collected and analyzed using a UV–visible spectrophotometer (Thermo Scientific GENESYS 10S UV-VIS, Madison, WI, USA) at a wavelength of 297 nm. All measurements were performed three times. The percentage encapsulation (%EE) and loading capacity (%LC) were determined using the equations mentioned below.
(2)$$\%\mathrm{EE}=\frac{{\mathrm{ALP}}_{\mathrm{Total}\;}{-\;\mathrm{ALP}}_{\mathrm{Free}\;}}{{\mathrm{ALP}}_{\mathrm{Total}\;}}\;\times 100$$
(3)$$\%\mathrm{LC}=\frac{{\mathrm{ALP}}_{\mathrm{Total}\;}{-\;\mathrm{ALP}}_{\mathrm{Free}\;}}{\mathrm{Total}\;\mathrm{SLNs}}\;\times 100\;$$

#### 3.6.4. Surface Morphology

Scanning electron microscopy (JSM-6360 LV, JEOL, Tokyo, Japan) was used to visualize the surface morphology of the samples chosen based on the smallest particle size and high EE%. The lyophilized samples were mounted on carbon tape and sputter coated with a thin gold layer in a high vacuum evaporator using a gold sputter module (JFC-1100 fine coat ion sputter; JEOL, Tokyo, Japan). After that, the coated samples were scanned and photomicrographs were taken at a 30 kV acceleration voltage.

#### 3.6.5. In Vitro Release Studies

The drug release profiles of the prepared formulations were determined using a dialysis bag (molecular weight cut off: 12–14 kDa). Appropriate volumes of the samples (equivalent to 1 mg ALP) were placed in the dialysis bag. The bags were immersed in 20 mL receptor fluid (PBS with 0.2% *w*/*v* Tween 80, pH 7.4) to confirm sink condition and stirred at 100 rpm with a shaking water bath at 37 °C. At specific intervals (0.15, 0.5, 1, 2, 4, 6, 8, 12, 24, and 48 h), 1.0 mL of receptor fluid was removed and replaced with the same volume of fresh buffer. After that, the withdrawn samples were centrifuged at 10,000 rpm for 15 min and the amount of released ALP extract in the supernatant was measured spectrophotometrically at a wavelength of 297 nm (Thermo Scientific GENESYS 10S UV-VIS, Madison, WI, USA). Then, a release profile was created based on a calibration curve. Furthermore, the release mechanism was identified from the released data after fitting with different release kinetic models (zero-order, first-order, Higuchi, and Korsmeyer–Peppas). The experiments were performed in triplicate.

#### 3.6.6. Antioxidant Activity

The ability of different ALP-loaded formulations to scavenge 2,2-diphenyl-1-picrilhidrazil (DPPH) can be used to assess their total antioxidative activity [41,42]. Different concentrations of ALP extract, ALP-SLN, C-ALP-SLN, and ALP-CSNP dispersions were made at 20 to 100 µg/mL using stock concentration (1 mg/mL). Then, 500 µL of each concentration was added to 125 µL of methanolic solution of DPPH (0.02%) and incubated for 1 h at room temperature in a dark place. The incubated samples were measured spectrophotometrically at 517 nm. The color shift from violet to colorless represented antioxidant activity. The solution of DPPH (free of ALP formulations and ALP solution) was used as a blank. Ascorbic acid was utilized as a positive control with the same procedure. All of the tests were performed in triplicate. The following equation was used to calculate the free radical-scavenging activity:(4)$$\%\;\mathrm{Free}\;\mathrm{radical}\text{-}\mathrm{scavenging}\;\mathrm{activity}=\frac{{\mathrm{Abs}}_{\mathrm{blank}}\;-\;{\mathrm{Abs}}_{\mathrm{sample}}}{{\mathrm{Abs}}_{\mathrm{blank}}}\;\times 100\;$$

A plot of the % free radical-scavenging activity of sample concentrations was used to estimate IC50 values.

#### 3.6.7. Antibacterial Activity

The antibacterial activity was screened firstly by microdilution broth assay according to the guidelines established by the Clinical and Laboratory Standards Institute (CLSI) with minor modifications [43]. The test was performed against bacteria such as *S. aureus* ATCC 29213 and *P. aeruginosa* ATCC 27853 (reference strains) for quality control. These bacteria were obtained from the American Type Culture Collection (Rockville, MD, USA). The stock solutions of the samples were diluted to obtain 200 µg/mL and 50 µL was added to 96-well plates. For inoculum preparation, 3 to 5 morphologically similar colonies from 24 h old culture were added to phosphate buffer saline and diluted to 0.5 McFarland standard (OD 0.08–0.1 at 625 wavelengths). This suspension was diluted 1:100 in media and 50 µL of the bacterial suspension was added to each plate to obtain the final concentration of 5 × 10^5^ CFU/mL. Then, the plates were incubated for 24 h at 37 °C. The growth of the bacteria was assessed by the color changing from yellow to red after the addition of 0.2 mg/mL p-iodonitrotetrazolium (INT). Each test was carried out in triplicate. The broth microdilution assay was utilized to detect the minimum inhibitory concentration (MIC) as described by the CLSI [43]. The stock solutions of the investigated samples (1000 µg/mL) were serially diluted two-fold in Mueller–Hinton broth (MHB) and 100 µL of each of these dilutions was added to 96-well plates to obtain concentrations ranging from 100 to 0.7 µg/mL. Bacterial strains were grown aerobically at 37 °C on the Mueller–Hinton agar. Vancomycin, penicillin, and streptomycin were used as reference drugs, and DMSO as control. Inoculum was prepared as described in the previous section, and 50 µL of standardized inoculum was added, resulting in an initial bacterial concentration of 10^5^ CFU mL^−1^, and incubated for 24 h at 37 °C. MIC values were expressed as the lowest concentration of the sample that inhibited bacterial growth completely. The bacterial inhibition was detected visually by the color changing from yellow to red and measured at 600 nm using µQuant (BioTek Instruments, Winooski, VT, USA). For all samples, a control plate without bacteria was prepared to subtract background absorbance. Each assay was performed in three independent tests in triplicate. The samples with MIC values ≤ 100 µg/mL were considered significantly active [44].

#### 3.6.8. Cytotoxicity Activity

The cell viability study of ALP extract and ALP-SLNs was evaluated by MTT assay using a variety of human cancer cell lines including lung carcinoma cells (A549), colon cancer cells (LoVo), and breast cancer cells (MCF-7). The cells were cultured in Dulbecco’s modified Eagle medium (Gibco, Grand Island, NY, USA) enriched with 10% fetal bovine serum (FBS) and 1% penicillin–streptomycin. The cells were preserved at 37 °C in a humidified incubator containing 5% CO_2_. Cells were seeded into 96-well plates at 5 × 10^4^ cells/well and allowed to adhere overnight. After that, the cells were treated with ALP extract and ALP-SLNs at concentrations ranging from 0 to 25 μg/mL for 48 h. Then, after aspirating the medium and washing the wells twice with PBS, 10 μL of MTT solution (5 mg/mL) was added to each well of the plate and incubated for an additional 4 h. The formed MTT–formazan crystal was dissolved in 100 μL of acidified isopropanol and the plates were inspected at 570 nm using a microplate reader (BioTek Instruments, Winooski, VT, USA) The following equation was used to determine cell viability.
(5)$$\%\;\mathrm{Cell}\;\mathrm{viability}\;=\frac{\mathrm{Absorbance}\;\mathrm{of}\;\mathrm{treated}\;\mathrm{cells}}{\mathrm{Absorbance}\;\mathrm{of}\;\mathrm{untreated}\;\mathrm{cells}}\times 100$$

The half inhibitory concentration (IC50) for each cell line was estimated from the dose–response curve using Origin software, version 8.

### 3.7. Statistical Data Analysis

The data were analyzed using the software packages Microsoft Excel 2010 and Origin software, version 8. Results are expressed as mean± standard error (*n* = 3). A one-way analysis of variance (ANOVA) was performed when comparing three or more conditions.

## 4. Conclusions

Cationic nanoparticles are considered an efficient technique used to enhance the biological activity of plant extract. Thus, ALP extract was encapsulated in CS-coated SLNs and CSNPs with favorable physicochemical properties. According to the results, SLNs could release ALP extract rapidly, while C-ALP-SLNs and CSNPs display sustained release. C-ALP-SLNs-F2 and ALP-CSNPs exhibited appreciable antioxidant activity. Particularly, C-ALP-SLNs and CSNPs containing ALP extract had considerably improved antibacterial activity compared to plain and uncoated ones. C-ALP-SLNs-F2 and ALP-CSNPs enhanced ALP’s antibacterial activity synergistically against *E. coli*, *S. aureus*, and *P. aeruginosa*. Interestingly, C-ALP-SLNs-F2 revealed significant cytotoxic activity against several cancer cell lines, including the A549, LoVo, and MCF-7 cell lines. These results indicate a promising therapeutic carrier, C-ALP-SLNs-F2, for use as an antimicrobial and antitumor agent.

## Figures and Tables

**Figure 1 molecules-28-03569-f001:**
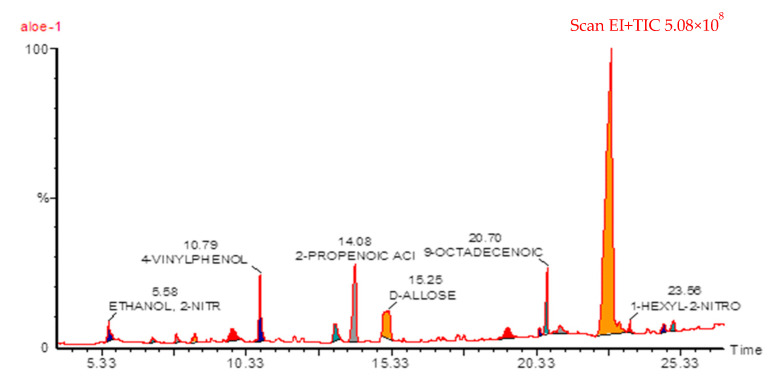
GC chromatogram of Aloe perryi extract.

**Figure 2 molecules-28-03569-f002:**
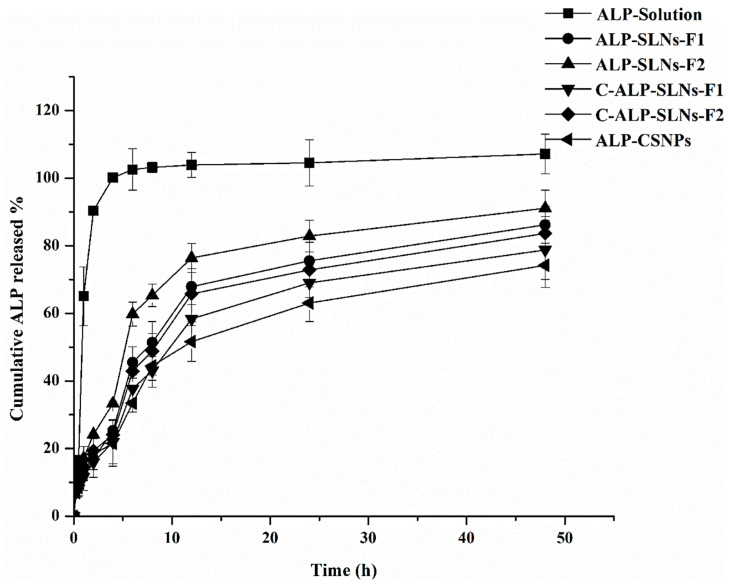
In vitro release pattern of ALP-loaded SLNs, C-SLNs, and CSNPs in PBS (pH 7.4) with 0.2% TW80.

**Figure 3 molecules-28-03569-f003:**
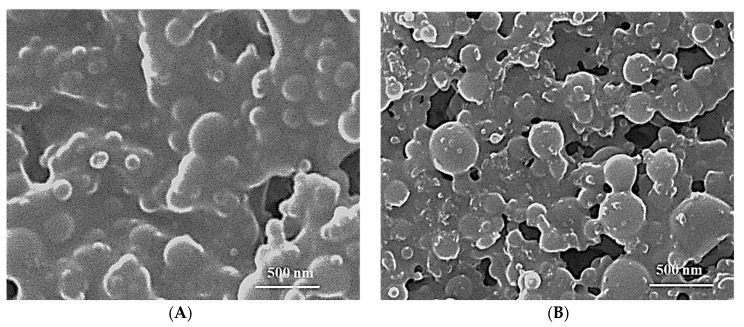
SEM images of SLN formulations: (**A**) ALP-SLNs-F2; (**B**) C-ALP-SLNs-F2.

**Figure 4 molecules-28-03569-f004:**
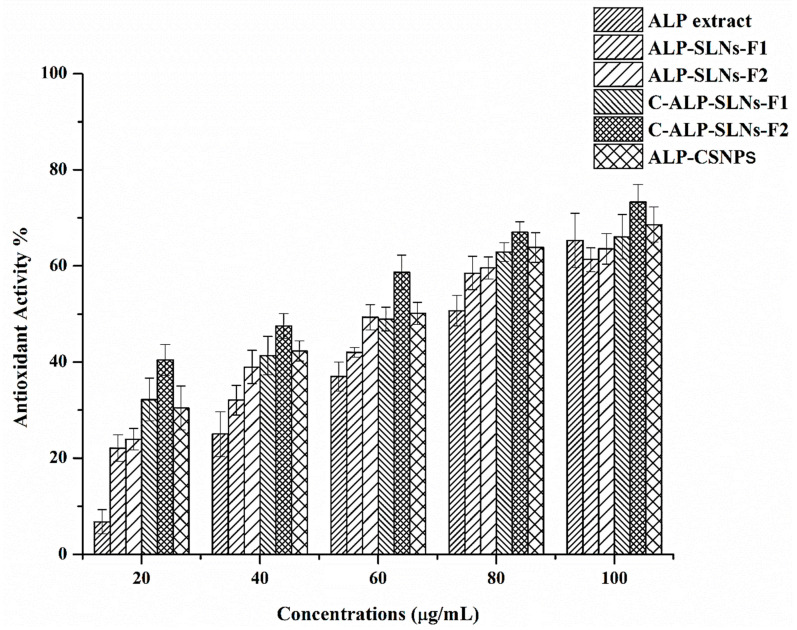
Antioxidant activity of ALP-loaded SLNs, C-SLNs, and CSNPs at different concentrations.

**Figure 5 molecules-28-03569-f005:**
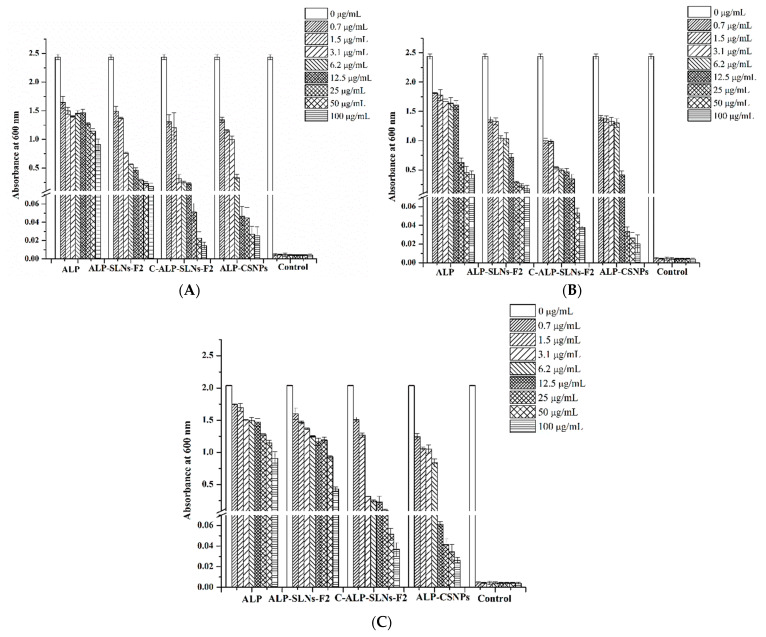
Antibacterial activity of ALP extract, ALP-SLNs-F2, C-ALP-SLNs-F2, ALP-CSNPs, and positive control against (**A**) *P. aeruginosa*, (**B**) *S. aureus*, and (**C**) *E. coli*.

**Figure 6 molecules-28-03569-f006:**
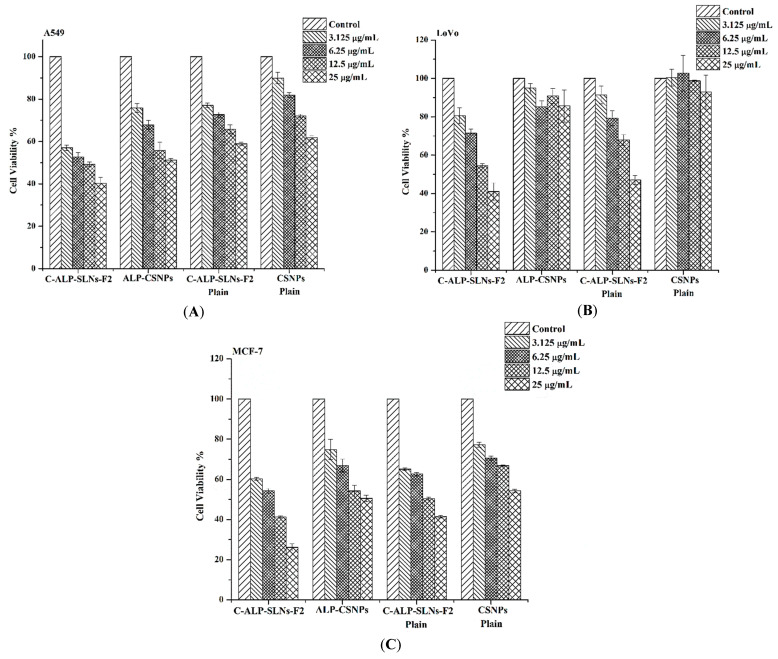
Cytotoxicity analysis of A549 (**A**), LoVo (**B**), and MCF-7 (**C**) cell lines treated with C-ALP-SLNs-F2 and ALP-CSNPs at various concentrations for 48 h.

**Table 1 molecules-28-03569-t001:** Major phytochemical compounds identified in *Aloe perryi*.

#	Name	RT	Ares %	Area
1	3-Amino-2-oxazolidinone	7.07	1.100	287506
2	N (2)-isobutyryl-2-Methylalaninamide	7.91	0.960	250712
3	Hydroxy dimethyl furanone	8.54	1.270	332917
4	2,3-Dihydroxy-propanal	9.86	6.240	1631520
5	4-Vinylphenol	10.79	10.470	2735098
6	5-Methyl-1,3-benzenediol	13.40	6.130	1602348
7	Cinnamic acid	14.08	10.030	2622387
8	d-allose	15.25	14.570	3807334
9	3-(4-hydroxy)-2-propenoic acid	19.38	2.600	679033
10	1,19-Eicosadiene	20.42	0.370	95965
11	9-Octadecenoic acid	20.70	3.440	899762
12	Caprinitrile	21.15	0.770	201889
13	9-Hexadecenoic acid	22.91	28.950	7566311
14	1-Hexyl-2-nitrocyclohexane	23.58	0.440	114156
15	Docos-13-enoic acid	24.72	0.690	181277

**Table 2 molecules-28-03569-t002:** The particle size, PDI, zeta potential, EE%, and DL% values.

Formulations	Particle Size (nm)	PDI	Zeta Potential (mV)	EE%	LC%
ALP-SLNs-F1	168.7 ± 3.1	0.18 ± 0.03	−12.4 ± 0.6	67.4 ± 2.4	3.53 ± 0.33
ALP-SLNs-F2	138.4 ± 9.4	0.17 ± 0.03	−15.8 ± 2.4	82.4 ± 1.7	5.18 ± 0.08
C-ALP-SLNs-F1	185.3 ± 5.5	0.20 ± 0.05	11.3 ± 1.4	64.7 ± 3.9	2.78 ± 0.14
C-ALP-SLNs-F2	173.6 ± 11.3	0.21 ± 0.02	13.6 ± 1.1	80.2 ± 5.3	4.66 ± 0.12
ALP-CSNPs	214.8 ± 6.6	0.26 ± 0.05	27.8 ± 3.4	76.4 ± 7.6	3.01 ± 0.29

**Table 3 molecules-28-03569-t003:** In vitro release kinetics models of SLNs, C-SLNs, and CSNPs (*n* = 3, mean ± SD).

Correlation Coefficient (R^2^)					
Formulations	Zero-Order	First-Order	Higuchi’s Model	Korsmeyer–Peppas Model	
	R^2^				*n*
ALP-SLNs-F1	0.847	0.983	0.948	0.959	0.419
ALP-SLNs-F2	0.799	0.985	0.921	0.943	0.373
C-ALP-SLNs-F1	0.904	0.980	0.983	0.992	0.401
C-ALP-SLNs-F2	0.905	0.987	0.982	0.993	0.360
ALP-CSNPs	0.848	0.984	0.951	0.996	0.382

**Table 4 molecules-28-03569-t004:** The physical storage stability of ALP-SLNs, C-ALP-SLNs, and ALP-CSNPs.

Formulations	SLNs Parameters	Zero Time	One Week	Four Weeks
ALP-SLNs-F2	Particle size (nm)	168.7 ± 3.1	170.3 ± 4.2	182.5 ± 3.9
	Zeta potential	−16.4 ± 0.6	−16.1 ± 0.7	−17.9 ± 0.8
	EE%	67.42 ± 2.4	66.23 ± 3.8	65.11 ± 4.1
ALP-SLNs-F2	Particle size (nm)	138.4 ± 9.4	141.5 ± 7.2	152.6 ± 6.1
	Zeta potential	−15.8 ± 2.4	−16.3 ± 2.8	−14.9 ± 3.3
	EE%	82.39 ± 1.7	84.65 ± 2.4	83.39 ± 2.7
C-ALP-SLNs-F1	Particle size (nm)	185.3 ± 5.5	189.1 ± 5.6	194.3 ± 4.7
	Zeta potential	11.3 ± 1.4	11.8 ± 2.3	14.2 ± 1.9
	EE%	64.65 ± 3.9	62.55 ± 4.1	63.21 ± 3.1
C-ALP-SLNs-F2	Particle size (nm)	173.6 ± 11.3	175.1 ± 11.3	181.4 ± 8.3
	Zeta potential	13.6 ± 1.1	14.2 ± 0.9	14.7 ± 1.8
	EE%	80.17 ± 5.3	78.22 ± 3.6	76.19 ± 5.2
ALP-CSNPs	Particle size (nm)	214.8 ± 6.6	221.5 ± 4.8	236.4 ± 8.5
	Zeta potential	27.8 ± 3.4	28.5 ± 4.2	28.3 ± 5.2
	EE%	78.83 ± 7.6	78.11 ± 7.6	79.06 ± 6.1

**Table 5 molecules-28-03569-t005:** Formulation composition of ALP-SLNs.

Composition	COMP (*w*/*w*%)	LP (*w*/*w*%)	TW80 (*w*/*w*%)	CS (mg/mL)
ALP-SLNs-F1	3.0	2.0	-	-
ALP-SLNs-F2	3.0	2.0	2.0	-
C-ALP-SLNs-F1	3.0	2.0	-	1.0
C-ALP-SLNs-F2	3.0	2.0	2.0	1.0
ALP-CSNPs	-	-	-	1.0

ALP-SLNs: solid lipid nanoparticles; C-ALP-SLNs: chitosan-coated solid lipid nanoparticles; ALP-CSNPs: chitosan nanoparticles; COMP: Compritol 888 ATO; PL: lipoid S100; TW80: Tween 80; CS: chitosan.

## Data Availability

The data presented in this work are available in the article.

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
