# Peer review of "Chitosan-Coated Solid Lipid Nanoparticles as an Efficient Avenue for Boosted Biological Activities of *Aloe perryi*: Antioxidant, Antibacterial, and Anticancer Potential"

_molecules, 2023, doi:10.3390/molecules28083569_

Round 1
Reviewer 1 Report
The manuscript by Tahany Saleh Aldayel et al. titled " Chitosan-coated solid lipids
nanoparticles as an efficient avenue for boosted biological activities of Aloe perryi: Antioxidant,
antibacterial, and anticancer potential" describes that the ALP-loaded nanosystems were
developed in order to increase their biological activity. As potential nanocarriers, solid lipid
nanoparticles (ALP-SLNs), chitosan nanoparticles (ALP-CSNPs), and SLNs coated with CS (CALP-
SLNs) were investigated. Overall, this is an interesting research question approach and a good
dataset has been provided. However, there are some points that must be treated before publication
consideration
My comments:
1. What is the primary issue that the study aims to address?
2. Abstract: Well written but can be improved with the incorporation of total results and
conclusions.
3. Do you think the subject is novel or current in the industry? Does it fill a particular hole
in the field?
4. Introduction: The introduction should have more information on methods and
detection techniques and suggested to include more latest references Please add the
below studies to your manuscript in the introduction section
https://doi.org/10.3390/molecules28062501
5. Use a unique description for the appliances used in experiments: name (company,
city, country)
6. Please make more discussions related to the data. It is important to correlate these data
with others - such as I also indicated in the introduction section. Overall, this part is just
presenting the data but discussions and comparisons with other papers would increase the
quality of the paper.
7. The quality of (Figure 3) should be improved
8. Conclusions: rewrite this section with more appropriate information with clarity
Author Response
Reviewer 1:
The manuscript by Tahany Saleh Aldayel et al. titled " Chitosan-coated solid lipids nanoparticles as an efficient avenue for boosted biological activities of Aloe perryi: Antioxidant, antibacterial, and anticancer potential" describes that the ALP-loaded nanosystems were developed in order to increase their biological activity. As potential nanocarriers, solid lipid nanoparticles (ALP-SLNs), chitosan nanoparticles (ALP-CSNPs), and SLNs coated with CS (CALP- SLNs) were investigated. Overall, this is an interesting research question approach and a good dataset has been provided. However, there are some points that must be treated before publication consideration.
Comment 1:
What is the primary issue that the study aims to address?
Author response:
Thank you very much for pointing out this comment. The primary issue of this study is to augment the biological activities of the aloe perryi and it was clarified in the last part of the introduction.
Aloe perryi has a special substance called C-glycosylated chromones, which is thought to be a phenolic compound and is the main reason for the high biological effect and hasn't been found in any other plant. Therefore, the current work's objective is to use CS-coated SLN and CSNPs as natural and safe carriers to enhance the effects of aloe perryi. The possible impact of this loading on the antioxidant, anti-microbial, and cytotoxic activities of aloe perryi was assessed. Aloe perryi in combination with solid lipid nanoparticles has not been previously reported in drug delivery.
Comment 2:
Abstract: Well written but can be improved with the incorporation of total results and conclusions.
Author response:
Thank you for this comment. The abstract was rewritten to incorporate the total results and conclusion in the revised manuscript.
Comment 3:
Do you think the subject is novel or current in the industry? Does it fill a particular hole in the field?
Author response:
Utilizing nanoparticles in conjunction with plant extract was introduced in recent years and some of the published works showed promising progress and solve problems associated with the aqueous solubility of plant extract.
Solid lipid nanoparticles have shown great potential in improving the effectiveness of various drugs and natural compounds. Therefore, the use of solid lipid nanoparticles as a delivery system for aloe perryi extract can significantly enhance its therapeutic benefits. Aloe perryi extract-loaded solid lipid nanoparticles have shown promising results for their biological activities. This approach has the potential to revolutionize the field of drug delivery by improving the biological activities of therapeutic agents. Further studies are needed to explore their potential in other therapeutic applications.
Comment 4:
Use a unique description for the appliances used in experiments: name (company, city, country)
Author response:
Thank you to the reviewer for this comment. A unique description of the devices was used.
Comment 5:
Introduction: The introduction should have more information on methods and detection techniques and suggested including more latest references Please add the below are studies to your manuscript in the introduction section https://doi.org/10.3390/molecules28062501
Author response:
As suggested by the reviewer, we have pointed out this reference in the introduction and references.
Comment 5:
Please make more discussions related to the data. It is important to correlate these data with others - such as I also indicated in the introduction section. Overall, this part is just presenting the data but discussions and comparisons with other papers would increase the quality of the paper.
Author response:
Thank you very much for your comment. The discussion was supported with other papers.
Comment 7:
The quality of (Figure 3) should be improved
Author response:
Thank you very much for your comment. The quality of Figure 3 was improved.
Comment 8:
Conclusions: rewrite this section with more appropriate information with clarity.
Author response:
Thank you for your remarks. The conclusion was rewritten.

Reviewer 2 Report
In this manuscript, authors developed ALP-loaded nano-22 systems to improve their biological activity with different nanocarriers where chitosan is applied. Comprehensive characterizations have been performed. In general, it is an interesting work and the manuscript is well-organized. However, there are still some issues to be addressed. A moderate revision is required before its acceptance.
1. The keyword characterizations can be removed or replace by other important words.
2. To have a better story line, it is better to divide the first paragraph into two or three paragraphs in introduction section.
3. Sentence structure issue “To the best of our knowledge, there is no one report….”
4. Why authors applied chitosan in this work should be further clarified with more detailed introduction on the structure, properties and applications of chitosan with necessary supporting articles: Sources, production and commercial applications of fungal chitosan: A review; Recent advancements in applications of chitosan-based biomaterials for skin tissue engineering; etc.
5. Simply introducing the strategy in this work can be applied at the last paragraph in introduction section.
6. The figure number A is missing in Fig. 5, also in fig. 6.
7. All the sub-images in one figure should be in the same page.
8. One sub-section to present the information of raw materials is required.
9. The equations should be numbered.
10. There are too many too old references, which is better to be deleted or replaced with recent articles to show the novelty of this work.
Author Response
Reviewer-2
In this manuscript, authors developed ALP-loaded nano-22 systems to improve their biological activity with different nanocarriers where chitosan is applied. Comprehensive characterizations have been performed. In general, it is an interesting work and the manuscript is well-organized. However, there are still some issues to be addressed. A moderate revision is required before its acceptance.
Comment 1:
The keyword characterizations can be removed or replace by other important words.
Author response:
As suggested by the reviewer, the keywords were modified.
Comment 2:
To have a better storyline, it is better to divide the first paragraph into two or three paragraphs in the introduction section.
Author response:
Thank you for the comments. The first paragraph was divided into two paragraphs in the introduction section.
Comment 3:
Sentence structure issue “To the best of our knowledge, there is no one report….”
Author response:
Thank you for the comments. The sentence structure was modified in the manuscript
Comment 4:
Why authors applied chitosan in this work should be further clarified with a more detailed introduction on the structure, properties, and applications of chitosan with necessary supporting articles: Sources, production and commercial applications of fungal chitosan: A review; Recent advancements in applications of chitosan-based biomaterials for skin tissue engineering; etc.
Author response:
Thanks for the reviewer’s comment. Some detailed information about chitosan was added to the manuscript
Comment 5:
Simply introducing the strategy in this work can be applied at the last paragraph in introduction section.
Author response:
Thank you for this appreciated comment. The strategy for this work was applied at the end of the introduction.
Comment 6:
The figure number A is missing in Fig. 5, also in fig. 6.
Author response:
Thanks for this comment. Figure number A, B, and C was added.
Comment 7:
All the sub-images in one figure should be in the same page.
Author response:
All the sub-images in one figure are now in the same page.
Comment 8:
One sub-section to present the information of raw materials is required.
Author response:
Thank you for this comment. The sub-section to present the information of raw materials was added.
Comment 9:
The equations should be numbered.
Author response:
Thank you for this comment. The equations were numbered.
Comment 10:
There are too many too old references, which is better to be deleted or replaced with recent articles to show the novelty of this work.
Author response:
Thank you for this comment. Some references were updated.

Round 2
Reviewer 1 Report
Comments and Suggestions for Authors
The authors made the suggested changes in the manuscript in response to my comments, and I believe the paper is now suitable for publication.
Author Response
First of all, thank you for your comments and suggestions that allowed us to greatly improve the quality of the manuscript.
Reviewer 2 Report
Authors have addressed most of the issues well except:
When generally introducing the chitosan, one recent important review article should be included: Sources, production and commercial applications of fungal chitosan: A review
Author Response
We appreciate the comment from the reviewer. As suggested by the reviewer, we introduce one of the recent important review on chitosan as a track change in the introduction.
CS is a natural polysaccharide obtained through the partial deacetylation of chitin. CS has many potentials uses in pharmaceutical fields due to its biosafety, biocompatibility, biodegradability, and biological activity [8]. Chitin is formed from N-acetyl-D-glucosamine units that are joined together by glycosidic (1,4) bonds to form the linear biopolymer [18]. It is the second most abundant polysaccharide in nature after cellulose, found in exoskeletons, cuticles, and fungi. The most common source of CS is from the shells of remaining crustaceans used in the seafood processing industry, such as crab or shrimp [18]. CS has a polycationic, harmless, non-toxic, and biodegradable chemical structure that is relatively stable, and it is biocompatible with a wide range of organs, tissues, and cells [18]. CS has been used in a variety of products, including food additives, wastewater treatment agents, live cell encapsulation, enzyme immobilization, cholesterol-lowering medications, biomedical wound dressings, and drug excipients in the pharmaceutical industry. Various factors, such as acid-base concentration, incubation time and temperature, and particle size have been found to affect the physicochemical properties of CS. Fungal CS has a lot of benefits for biomedical applications and wound dressings due to its molecular properties and wound-healing capabilities [18]. It has a lower antigen effect and poly-cationic properties that make it soluble in physiological pH ranges. It can be utilized as a non-viral gene delivery system and potential drug carrier. CS was used in wound dressings in two ways: initially. CS is highly effective against dangerous microorganisms at low concentrations while being less poisonous to mammalian cells than other compounds [18]. The products of CS have enormous economic potential in the biomedical/tissue engineering and food industries [18]. Therefore, numerous ways for functionalizing CS for these purposes have been developed. Unfortunately, chitosan's commercial use is still fairly limited. CS has been used as a coating material to generate numerous cationic nanoparticles, including SLNs in the field of pharmaceutical nanotechnology [19]. Remarkably, CS-coated SLNs (C-SLNs) are reportedly very promising carriers for the delivery of a variety of natural drugs [19].
Reference:
Tanzina, H.; Khan, A.; Brown, D.; Natasha, D.; Zhibin, H.; Yonghao, N. Sources, production and commercial applications of fungal chitosan: A review. J. Bioresour. Bioprod. 7 (2) 2022, 85-98.